# Research on plasma arc flame length detection technology based on region of interest

Jie Li[1,2], Wei Jiang[1]*, Jian Lei[1], Xiaoxiao Xing[1,2]

**1** Electronic Information and Electrical College of Engineering, ShangLuo University, Shangluo, Shaanxi, China, **2** Artificial Intelligence Research Center of Shangluo, Shangluo, Shaanxi, China

* jw304059493@163.com

## Abstract

With the rapid advancement of metal 3D printing technology, there is a growing demand for spherical metal powder as a primary material for 3D printing. The process technology that ensures the production of high-quality spherical metal powder has become a focal area of research for numerous enterprises and research institutions globally. In the conventional plasma rotating electrode method for powder production, the feed speed of the servo feeding mechanism is manually predetermined, leading to potential variations in the distance between the end face of the metal rod and the plasma gun that generates the plasma arc. Such inconsistency can compromise the quality of the metal powder produced and pose safety hazards if the gap between the metal rod and the plasma gun is too narrow. To address these issues, this study presents a novel plasma arc length detection system based on the concept of the region of interest. The proposed system leverages image processing technology for efficiently detecting the plasma arc length. By incorporating image detection within the region of interest alongside an arc length correction function, the system enhances real-time performance and detection precision. Additionally, real-time monitoring of the detection site is enabled through KingView. Experimental findings indicate that the image target area post plasma arc detection exhibits well-defined edges, clear brightness, and minimal noise, thereby meeting the prerequisites for subsequent image processing and monitoring tasks. The corrected plasma arc length averages around 40mm, with a detection error of less than 1mm when compared to the desired controlled plasma arc length. Moreover, the length variation remains relatively stable, thus fulfilling the measurement criteria. Over time, the detected plasma arc length exhibits negligible fluctuations, suggesting consistent proximity between the plasma gun and the end face of the metal rod during the melting process. The controller can dynamically control the feed speed of the servo feeding mechanism according to the detected plasma arc length, ensuring a constant distance between the plasma arc and the end face of the metal rod throughout the powder production process, thus aligning with practical industrial requirements.

**Data availability statement:** All relevant data are within the manuscript and its Supporting Information files.

**Funding:** This work was supported by the Shaanxi Provincial Department of Education's General Special Scientific Research Project (23JK0417), the Shaanxi Provincial Department of Education's General Special Scientific Research Project (24JK0419), the Shangluo University Natural Science Research Project (22SKY002), and the Qin Chuangyuan Cites High-level Innovation and Entrepreneurship Talent Programs of Shaanxi Province (Project No. QCYRCXM-2022-367).The funders had no role in study design, data collection and analysis, decision to publish, or preparation of the manuscript.

**Competing interests:** The authors have declared that no competing interests exist.

## 1 . Introduction

As the global manufacturing sector upgrades, manufacturing is growing faster than ever. Notably, 3D printing technology [1] stands as a representative of advanced manufacturing techniques that have significantly impacted the development of the manufacturing industry. In the realm of 3D printing, metal powder serves as a key material [2]. The main methods for preparing metal powders include plasma rotating electrode preparation (PREP), plasma atomization (PA), and aerosol gas atomization (GA) techniques [3]. Among these methods, the plasma rotating electrode approach is highlighted for its ability to produce spherical metal powder with minimal oxygen content and high cleanliness [4]. In the plasma rotating electrode method, the process of preparing metal powders is typically carried out in a controlled gas environment of high-purity argon to prevent oxidation and impurities. Despite its advantages in producing high-quality metal powders, challenges such as empirical setting of the metal rod feed speed and the instability of the plasma arc can impact the consistency and safety of the process. Maintaining a constant distance between the metal rod and the plasma gun to ensure optimal powder quality is crucial, as deviations in this distance can lead to safety hazards and operational issues, such as explosions or back spray incidents.

Extensive research has been dedicated to enhancing flame detection technologies. Jiang B et al. [5] proposed a method that combines local texture features and global color features of the LAB histogram to achieve swift and accurate flame detection. Similarly, another study [6] utilizes a classifier to pinpoint the flame's target area through the extraction of texture features in the flame region. With the rapid evolution of deep learning technologies, neural networks have also found application in the sphere of flame detection. Frizzi et al. [7] introduced a flame detection approach based on convolutional neural networks, which exhibits effective detection capabilities; however, the detection accuracy varies significantly when encountering flames of different colors. Muhammad et al. [8] presented a flame identification method utilizing the GoogleNet model [9]. This method incorporates transfer learning to fine-tune model parameters, striking a balance between detection efficiency and accuracy, albeit resulting in a reduction in detection accuracy. In a separate document [10], a video-based flame recognition method was proposed. By employing multi-scale fusion to extract flame features, the detection rate was enhanced, significantly boosting real-time performance. Furthermore, a study [11] introduced a flame identification algorithm based on an enhanced version of the YOLOv5 network [12]. Leveraging transfer learning [13], this method enhances the small target detection layer, augments the model's correlation identification capacity, and ultimately improves flame detection accuracy. While these methods have shown promise in detecting parameters such as flame shape, category, and area, they fall short in capturing length information of the flame.

Some researchers have explored the detection of plasma arc in various studies. A noteworthy piece of literature [14] employs spectroscopy as a technique to assess plasma arc, utilizing blackbody radiation to analyze prominent spectral lines and determine the temperature of the plasma arc. Prasad et al. [15] introduced a simulation-based approach to quantify the plasma current within the International Thermonuclear Experimental Reactor (ITER). Their method involved investigating the polarization rotation induced by Faraday effect in a rotating fiber optic sensor situated around the vacuum chamber, accounting for bending and twisting effects to assess the reflectometer's performance in gauging plasma current in the ITER facility. Another scholarly work [16] utilized a pair of synchronized high-speed cameras to capture the dynamics of both the plasma column and tracer particles. This study conducted a comprehensive 3D data analysis of the column and tracer particles, encompassing the reconstruction of the plasma column in three dimensions and the measurement of tracer particles' velocities

using discrete tomography. By determining the 3D slip velocity and length of the plasma column, researchers were able to accurately estimate the radius of the conductive zone within the plasma column. While existing research has focused on measuring parameters such as plasma arc temperature, density, and conductivity, investigations specifically targeting the measurement of plasma arc length remain limited.

Fig 1 provides a visual representation of the rationale behind the plasma arc flame length detection process and its critical role in our operations. In the initial stages of the metal pulverization procedure, a plasma gun injects high-temperature plasma into a metal rod, causing the surface of the metal rod to melt under the intense heat of the plasma arc and the centrifugal force of droplets. Subsequently, the molten metal undergoes condensation, resulting in the formation of metal powder. During this process, precise control of the plasma arc is essential as any fluctuations can jeopardize the consistency of the metal powder and potentially lead to safety hazards. To address these challenges, we have devised a metal powder device that incorporates a plasma arc flame length detection system. This system replaces the conventional control rod feeding mechanism with a design that regulates the plasma gun feed based on real-time measurements of the plasma arc flame length. By utilizing servo mechanisms to adjust the speed of metal rod transverse feed according to plasma arc fluctuations, we can maintain a constant distance between the metal rod end surface and the plasma gun. This optimization not only enhances the quality of the produced metal powder but also mitigates safety risks associated with unstable plasma arcs. Plasma arcs, characterized by their extreme temperatures and intense luminosity, present unique challenges for length detection, necessitating specialized methodologies. In pursuit of this goal, the present study focuses on Plasma Arc Flame Length Detection Technology [17]. By integrating image processing techniques into plasma arc length measurement [18], we have developed a novel detection algorithm that defines the region of interest within the plasma arc flame and ensures accurate measurements in real-time. Importantly, our non-intrusive detection approach enhances safety and reliability by eliminating the need to physically probe the plasma arc, thereby offering significant advancements in plasma detection methodologies.

The article is structured into four main sections as outlined below:

The first section is the Introduction, which provides a concise overview of the content and significance of the paper. It evaluates the current status of flame detection technology and

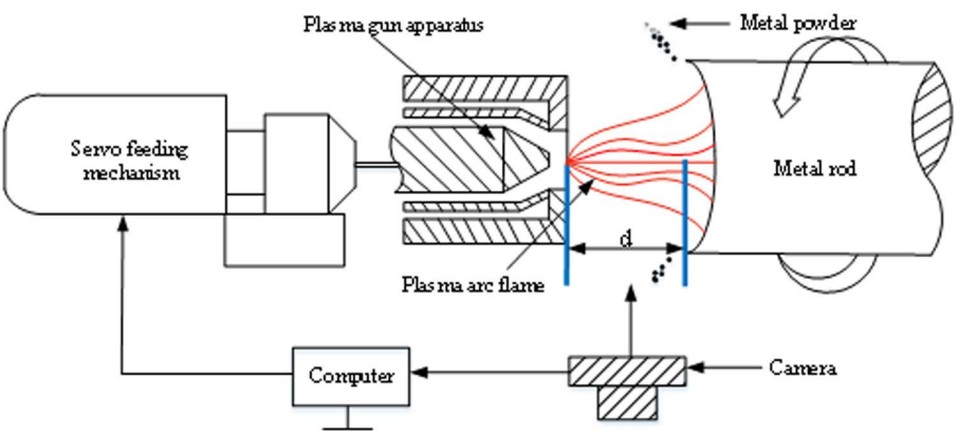

**Fig 1. Metal Pulverizing Device Based on Detection of Plasma Arc Flame Length Detection.**

plasma detection technology, and succinctly outlines the rationale and functionality of the plasma arc flame length detection system developed for this study.

The second section focuses on the design of the plasma arc flame length detection system. The architecture is tailored to meet the specific requirements of plasma arc flame detection, with a defined acquisition system layout. A system centered around determining the plasma arc flame length based on the region of interest is engineered to enable real-time monitoring of flame length.

The third section analyzes the experimental results. The focus shifts to the evaluation of experimental outcomes concerning the quality and precision of plasma arc flame length measurements. The analysis delves into practical observations regarding the absence of accurate plasma arc flame length readings, validating the effectiveness of the detection system in real-world industrial settings.

The fourth section is the Conclusion. A summary of the work conducted is provided, along with a glimpse into the future research trajectory to guide subsequent investigations.

## 2. Design of plasma arc flame length detection system based on region of interest

The demand for detecting plasma arc flame length in this study comprises several crucial components. Firstly, there is a need to capture the video signal of the plasma arc in real-time and accurately determine the precise position and length of the plasma arc flame. Subsequently, the identified video signal and length of the plasma arc flame should be promptly transmitted to the KingView software for the servo feed system to access the information instantaneously [19]. Finally, these data must be presented in the KingView interface with the flexibility to adjust detection parameters such as binary parameters, brightness, and filter width [20]. Consequently, the holistic framework of the plasma arc flame detection system is structured into three main segments: the acquisition system, detection system, and display system, illustrated in Fig 2.

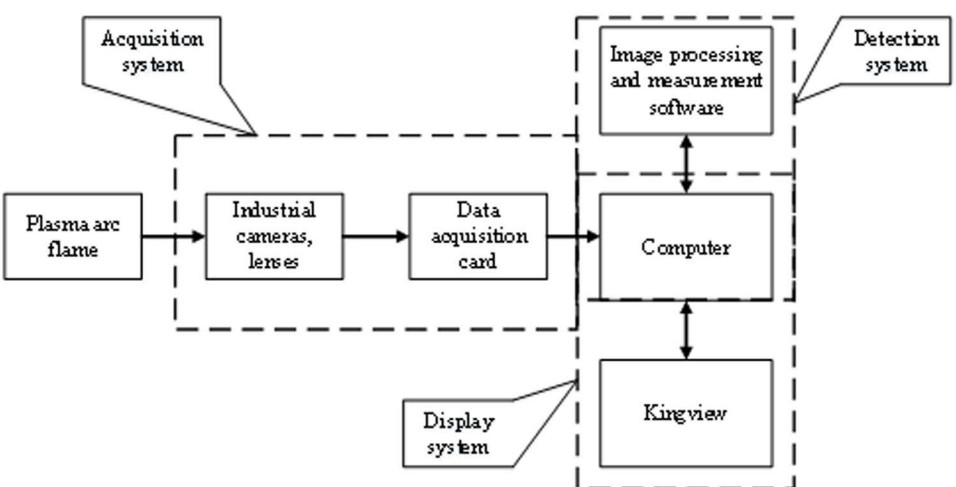

**Fig 2. Overall Architecture Design of Plasma Arc Flame Detection System.**

## 2.1 Design of the image acquisition system

The acquisition system is mainly composed of industrial cameras, lenses, and data acquisition cards. The specific requirements for this detection task are outlined as follows: (1) Working distance: 1.5 meters; (2) Detection field of view: 80 millimeters; (3) Detection accuracy: 0.1 millimeters; (4) Detection speed: under 0.1 second per frame, corresponding to approximately 10 frames per second for the camera.

The selection of the camera and lens is guided by the plasma arc flame detection criteria. Camera selection primarily hinges on resolution and speed considerations. The resolution calculation, based on the detection accuracy and field of view, dictates the need for a 3-million-pixel camera, ensuring an appropriate balance between pixel count and detection accuracy. Insufficient pixel count would compromise accuracy, while excessive pixels would impede processing speed, failing to meet real-time constraints. Given that flame imaging entails motion capture, a camera boasting global exposure capabilities and a speed of 30 frames per second is recommended.

Lens selection is contingent upon magnification and focal length requirements. The desired lens magnification should adhere to the formula: camera pixel count * vertical resolution/ detection field of view = 0.12. Additionally, the focal length of the lens is determined by: working distance * (magnification/ 1 + magnification) = 134 millimeters. A tabulated overview of the acquisition system components is presented in Table 1 for reference.

## 2.2 Design of image detection system

The recognition system is the most critical part of this system. It is responsible for processing the collected plasma arc flame images, accurately detecting the length of the plasma arc flame, and sending the processed images and the detected plasma arc flame length data into the KingView software. The process of the detection system designed in this article is shown in Fig 3.

The processes outlined in the detection system depicted in Fig 3 correspond to the functionalities of the image processing and measurement software featured in Fig 2. Subsequent to the actions of the acquisition system in Fig 2, the system secures a color image of the plasma arc flame. As indicated in Fig 3, the color image is typically initial transitioned into a grayscale format [21], following which the grayscale images undergo processing. Despite passing through the capture card, the images captured by industrial cameras often retain

Table 1. Data Acquisition System Equipment.

| Equipment | Model number | Parameters |
|---|---|---|
| Camera | BFLY-PGE-31S4C-C | 3.2MP Col. BFLY-PGE CMOS C-Mount |
| Lens | FA7501C | 75mm Focal length, 5MP Resolution |
| Data cable | MV-1-1-3-5M | Super Category 6 Gigabit cable, 5M |
| Power cord | 06PF-BF-A-5M | With Blackfly camera, Tape triggering, 5M |
| Acquisition card | GIGE-PCIE2-2P02 | Two-channel, PCIEx2, POE |

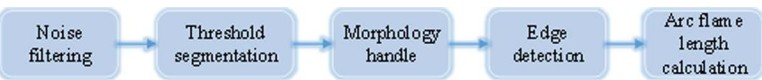

Fig 3. Detection System Process.

a considerable amount of noise [22]. Removing these noise elements is imperative for the detection goals. In this study, the approach involves employing median filtering [23] to eliminate noise within the image. Median filtering proves highly effective in eliminating discrete noise points within the image. For each given area, the algorithm selects a pixel point within that region. The median pixel value is then designated as the value of the central point, effectively eliminating point noise and preserving the sharpness of image edges, thereby enhancing overall image fidelity. To detect the plasma arc flame in the collected image, it is necessary to remove the background, extract the target flame area, and perform threshold segmentation on the image [24] to separate the target area from the background. Threshold segmentation involves determining a suitable threshold level to classify each image point as either background or target, hence delineating the target region. The threshold determination methodology is experimentally based, with the study ultimately selecting a threshold value of 200 after numerous trials to achieve optimal segmentation results. As the plasma arc flame tends to emit high brightness, small holes may persist post-segmentation. Addressing these holes is crucial to accurate plasma arc flame length calculations. Leveraging morphological operations [25], these holes can be effectively filtered out. Through multiple experiments, the study opts for a 5*5 square structural element for opening the image post-threshold segmentation. This method involves erosion followed by dilation, preserving the target area's shape while removing smaller holes without compromising the image condition. Complying with the plasma arc flame detection requirements necessitates extracting the edges within the target area. Consequently, conducting edge detection on the target area is pivotal. This study uses the canny algorithm [26] for edge detection, utilizing dual thresholds [27] to ascertain genuine and potential edges within the target region. Notably, the advantage of this algorithm is its resistance to noise interference, with the extracted edges demonstrating impressive continuity while avoiding false positives or edge loss.

The calculation of length becomes feasible subsequent to the extraction of the boundary of the plasma arc flame. Image processing methodology enables the acquisition of the positional data for both the leading and trailing edges of the plasma arc flame: the left point $x_{left}(i_1, j_2)$ and the right point $x_{right}(i_2, j_2)$. By virtue of this technology, the pixel length of the plasma arc flame can be accurately determined utilizing the following formula (1):

$$l = j_2 - j_2 \tag{1}$$

To carry out this procedure, a ruler is positioned adjacently to the plasma arc flame, while maintaining a constant camera angle. Subsequently, an image of the ruler is captured and subjected to the same pre-processing steps detailed previously. Through the application of formula (1), the pixel length of the ruler can be computed and denoted as N. The known physical length of the ruler, as measured directly with a measuring device, is represented as y. Thus, the real-world length of a single pixel within the ruler image is expressed in formula (2):

$$c = \frac{y}{N} \tag{2}$$

Given that the distance between the camera and the plasma arc flame is equivalent to the distance between the camera and the ruler, the physical length of the pixel unit on the ruler can be considered as the true length of the pixel unit on the plasma arc. Consequently, the accurate measurement of the plasma arc's length can be derived using the following formula (3):

$$d = l^{\star} c \tag{3}$$

In the actual detection process, the position of the camera may affect the accuracy of the plasma arc flame length detection. Therefore, it is necessary to correct the initially detected length. The actual length of the detected plasma arc flame after correction is shown in formula (4):

$$d' = d * k \qquad (4)$$

where K is the correction factor, which can be represented by formula(5):

$$k = \frac{d_1}{d_2} \qquad (5)$$

where
$d_1$ is used to measure the distance between the plasma gun and the end face of the metal rod as measured by the measuring tool, which is also the actual distance of the plasma arc; and.
$d_2$ represents the length of the plasma arc flame detected before correction.

## 2.3  Image detection based on the region of interest

The quantity of pixel points in every frame image acquired by the camera is substantial. To uphold detection precision, the image must not undergo compression, thereby escalating the computational load in image processing and impeding real-time detection performance. Consequently, this study devises an image detection approach centered on the region of interest, illustrated in Fig 4b.

In Fig 4(a), the absence of a defined region of interest necessitates processing the entire image to determine the length of the plasma arc flame, leading to increased computational requirements and a subsequent reduction in image processing speed. Real-time detection for video images becomes challenging under such circumstances. Contrastingly, Fig 4(b) demonstrates the implementation of a region of interest. Typically, the region of interest is defined as a 500 * 500-pixel area in the image. By focusing the image processing solely on this designated area, significant enhancements in processing efficiency are achieved. Furthermore, the intense brightness exhibited by the plasma arc flame often results in the generation of large noise artifacts around the image, which are complex to eliminate using conventional methods. By establishing a predetermined region of interest, the necessity to process larger noise elements

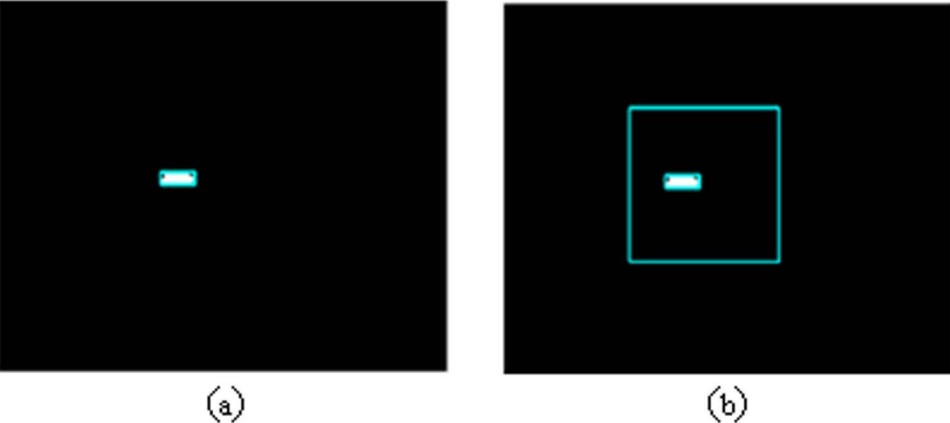

**Fig 4.  Image Detection Based on the Region of Interest.**

beyond this area is obviated. Notwithstanding, it is possible that sizable noise artifacts persist within the designated region. To address this issue, this study introduces a novel noise filtering technique predicated on the utilization of an area threshold. This methodology involves quantifying the area of all enclosed regions within the image, setting a specific area threshold, and subsequently replacing closed areas with an area below this threshold with a complementary background fill.

## 3 Experimental results and analysis

### 3.1 Results and analysis of image detection quality

The accuracy of image detection significantly influences the subsequent determination of plasma arc length. A robust image detection process should exhibit consistent characteristics: absence of voids in the target area, well-defined edges, and minimal interference with extraneous brightness levels. Through a series of empirical investigations, it was observed that the efficacy of image detection is predominantly contingent upon the threshold setting. Despite the introduction of a light-shielding mirror in front of the lens, the plasma arc image remained excessively illuminated. Optimal threshold selection is pivotal for attaining precise detection outcomes. This paper aims to delve into the repercussions of threshold values on image detection quality. An array of threshold values ranging from 170 to 210 are evaluated against the plasma arc flame detection quality effect map, as illustrated in Fig 5.

As illustrated in Fig 5, the choice of a threshold of 170 renders the morphology of the plasma arc indiscernible. This is attributed to the intense luminosity of the plasma arc flame which hampers effective separation from the background due to the inadequacy of a low threshold value. Increasing the threshold to 180 offers a somewhat clearer depiction of the plasma arc's form, yet residual areas of adhesion and smaller voids persist. Elevating the threshold beyond 190 facilitates the differentiation of the plasma arc's shape; however, the

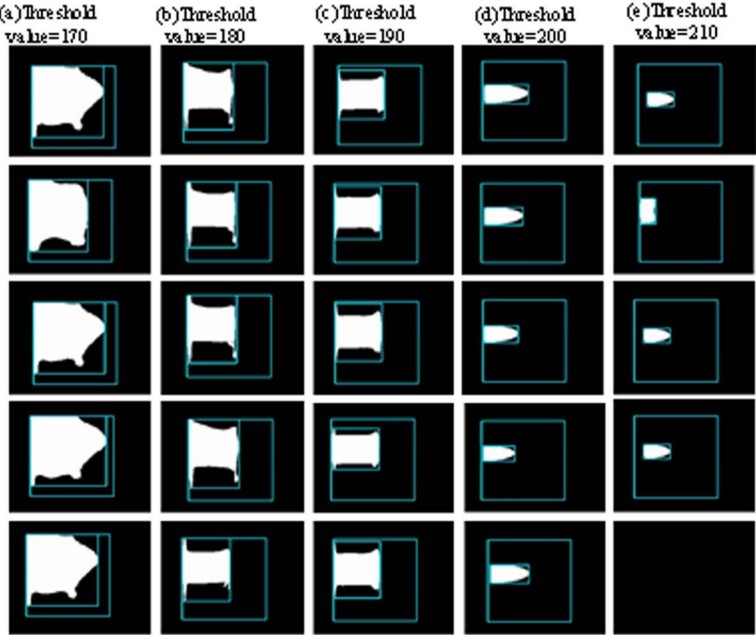

**Fig 5. Plasma Arc Flame Detection Effect.**

delineated edges appear coarse with vertical protrusions at both terminations of the arc flame. Upon setting the threshold at 200, distinct isolation of the plasma arc flame from the background is achieved, resulting in smoother image contours. Further elevations in the threshold lead to restricted detection solely at the central position of the plasma arc flame, which proves erratic, impeding accurate plasma observation. These ramifications pose a challenge to the subsequent measurement of the plasma arc flame's length. Progressing with heightened thresholds eventually culminates in the complete inscrutability of the plasma arc's morphology. Consequently, this study opts for a threshold value of 200, a decision validated by the detection outcomes of the plasma arc flame, as depicted in Fig 6.

The original image of the plasma arc flame captured is depicted in Fig 6(a), while Fig 6(b) illustrates the plasma arc flame detection image obtained at a threshold of 200. The identified target area's boundary within the plasma arc flame appears well-defined with a smooth edge, exhibiting a clear and noise-free brightness level. This ensures suitability for requisite image processing and monitoring tasks.

## 3.2 Results and analysis of plasma arc flame length detection accuracy

The accurate detection of the plasma arc length is essential for effectively controlling the feed speed of the servo feeding mechanism. Any inaccuracies in measuring the plasma arc flame length can compromise the subsequent control of the servo feeding mechanism. The criterion for precision in plasma arc flame length detection in this research is an error margin of less than 1 mm. The distance between the plasma gun and the metal rod's end face to be controlled is 40 mm, remaining consistent throughout the experiment's initial stages. Consequently, the plasma gun should maintain a 40 mm distance from the metal rod's end face. The ideal length of the detected plasma arc flame falls within the range of 39 mm to 41 mm, as deviations outside this range can disrupt the servo feeding mechanism's control.

There are many factors that affect the detection of plasma arc flame length, such as whether the detection of the plasma arc image is accurate in extracting the target area, whether the distance from the camera to the plasma arc is equal to the distance from the camera to the ruler, and whether the angle between the camera shooting the plasma arc and the shooting ruler, and whether the angle of the camera shooting the plasma arc is the same as the angle of the shooting scale. The configuration of the plasma arc flame is significantly impacted by the threshold value, posing challenges in establishing correspondence between the identified target area and the scale. Moreover, subjective factors come into play with regards to the angle at which the camera captures the plasma arc flame. Consequently, this study exclusively examines the impact of camera-to-plasma-arc distance on detection accuracy, with specific

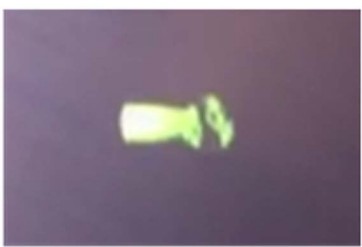
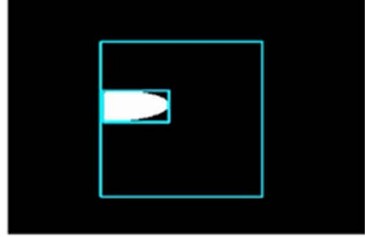

(a) Plasma arc flame original          (b) Plasma arc flame detection diagram

**Fig 6. Plasma arc flame detection results.**

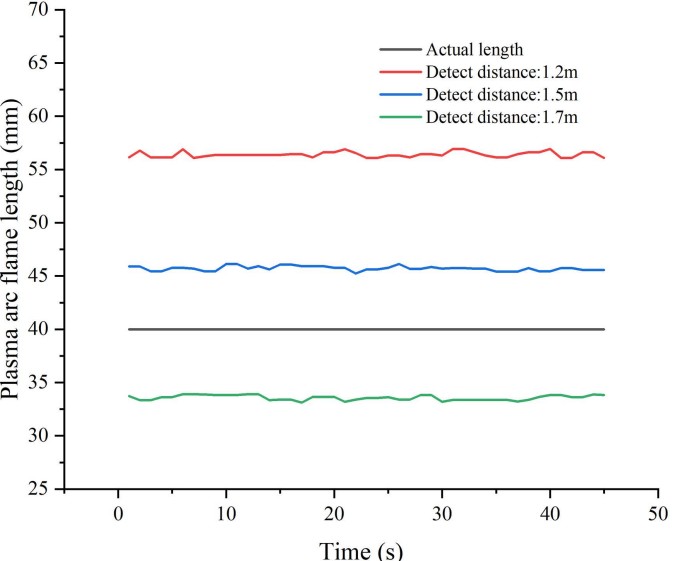

**Fig 7. Plasma Arc Flame Length at Different Detection Distances.**

distances set at 1.2 m, 1.5 m, and 1.7 m. The resulting lengths of the detected plasma arc flames are depicted in Fig 7.

Fig 7 illustrates that at a camera-to-plasma arc flame distance of 1.2 meters, the measured length is approximately 56mm, deviating notably from the required 40mm. This discrepancy is primarily attributed to the wider field of view at shorter distances, resulting in an exaggerated pixel length for the detected plasma arc flame and consequently large detection errors. At a detection distance of 1.5 meters, equivalent to the camera-to-ruler distance, the measured length remains inadequate at approximately 45mm, still falling short of the desired 40mm. This discrepancy is primarily due to the intense brightness and distinctive shape of the plasma arc flame. The challenge lies in accurately delineating the aperture layer within the image area, leading to an overestimation of the detected pixel length of the plasma arc flame.

At a detection distance of 1.7 meters, the measured length decreases to about 33mm. The larger far field of view results in a reduced pixel length of the detected plasma arc flame, which gives rise to the noted reduction in detection length and an escalation of detection inaccuracy.

The experimental observations underscore the substantial influence of subjective biases on detection accuracy, posing challenges in their elimination. As a remedy, a correction function is proposed in Formula (4) to rectify the measured plasma arc flame length. Fig 8 portrays the corrected results, showcasing the efficacy of the formula in refining the detected length.

The correction process is completed within the detection system, and there is no need to output the results for correction later. As can be seen from Fig 7, the corrected plasma arc flame length is about 40mm. When compared with the plasma arc flame length to be controlled, the margin of detection error is below 1mm, demonstrating a consistent and stable length variation. This level of precision fulfills the practical requirements of industrial production as outlined in this study.

### 3.3 Control effect analysis

The purpose of this study is to ensure that the distance between the plasma gun and the end face of the metal rod is constant during the process of melting the metal rod. Directly

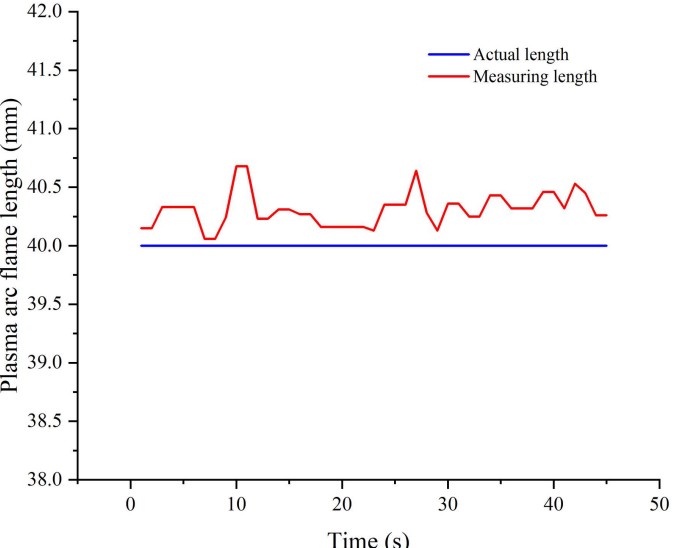

**Fig 8. Corrected Plasma Arc Flame Length.**

measuring this distance using conventional tools is challenging during the melting procedure. However, by observing the extent of the plasma arc flame during detection, the distance between the plasma gun and the end face of the metal rod can be ascertained. In this study, the length of the plasma arc flame measured serves as a proxy for this distance, enabling the evaluation of its stability. Variations in the length of the plasma arc flame at distinct time intervals can be analyzed to determine the constancy of the distance between the plasma gun and the end face of the metal rod. Fig 9 illustrates the lengths of the plasma arc flames detected during three separate intervals within the same experimental setup, each spaced 20 minutes apart.

In the course of the experiment, the length of the plasma arc flame was consistently measured at various intervals. It can be seen that the length of the plasma arc flame remained relatively constant over time, consistently hovering around 40 mm, a distance prescribed between the plasma gun and the terminal point of the metal rod. This distance exhibited negligible variations throughout the duration of the experiment. Significantly, the metal rod underwent continuous melting throughout the experimental procedure, elucidating the controller's dynamic capacity to regulate the feed rate of the servo feeding mechanism in real-time. This regulation is facilitated by the controller's ability to adapt the servo feeding mechanism based on instantaneous measurements of the plasma arc flame's length. Consequently, this ensures a consistent distance between the plasma arc flame and the end face of the metal rod during the powder manufacturing process, thereby meeting the requisite control standards.

## 4. Conclusions

In traditional plasma rotating electrode pulverizing processes, a common challenge is the inability to maintain a constant distance between the rod end face and the plasma gun. To address this issue, this study introduces a novel plasma arc flame length detection technology based on the region of interest. Leveraging image processing techniques, we have developed a specialized algorithm for detecting and correcting the plasma arc flame length within the region of interest, enabling real-time monitoring of this parameter. By continuously adjusting the feed speed of the servo feed mechanism within the metal pulverizing device based on the

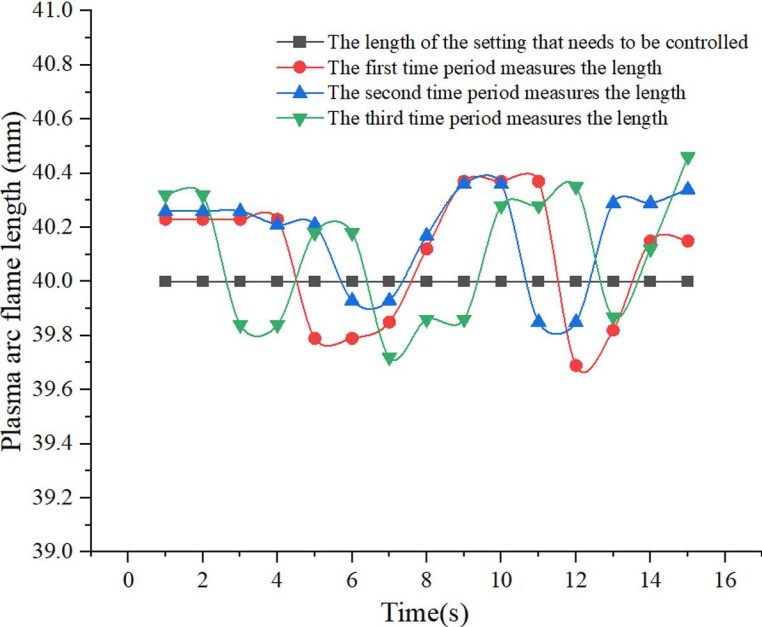

**Fig 9. Plasma Arc Flame Length at Different Periods.**

detected plasma arc flame length, we ensure a consistent distance between the plasma gun and the metal rod's end face. This not only enhances pulverizing quality but also mitigates safety risks. Experimental results demonstrate that the corrected plasma arc flame length averages approximately 40mm, with a detection error of less than 1mm compared to the targeted length. This accuracy level meets the stringent requirements of industrial production. Furthermore, an analysis of the servo feed mechanism's control efficacy reveals that the integrated plasma arc flame detection and servo feed system effectively fulfill the control objectives. Given the constraints imposed by the experimental apparatus and the state of progress in the powder fabrication process outlined in this study, the subsequent phase entails the examination of the spatial separation between the metallic rod and the plasma gun. Furthermore, the utilization of an improved detection algorithm to accurately measure the length of the plasma arc flame will be pursued.

## Supporting information

**S1 Raw data. These excel spreadsheets contain the raw data from the experiments that were analyzed in this article.** It is separated by subject as well as by measurement type and measurement number.
(ZIP)

## Author contributions

**Conceptualization:** Jie Li, Jian Lei.

**Data curation:** Jie Li.

**Funding acquisition:** Wei Jiang.

**Methodology:** Jie Li.

**Project administration:** Jie Li.

**Software:** Jie Li, Xiaoxiao Xing.

**Validation:** Jian Lei, Xiaoxiao Xing.

**Writing – original draft:** Jie Li.

**Writing – review & editing:** Jie Li, Jian Lei.

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
