## [Decision Letter · Decision Letter 0]

9 Jul 2024

PONE-D-24-07571Research on Plasma Arc Flame Length Detection Technology Based on Region of InterestPLOS ONE

Dear Dr. Li,

Thank you for submitting your manuscript to PLOS ONE. After careful consideration, we feel that it has merit but does not fully meet PLOS ONE’s publication criteria as it currently stands. Therefore, we invite you to submit a revised version of the manuscript that addresses the points raised during the review process.

The presented work has been found commendable for its direct utility in a practical scenario. However, the reviewers have raised concerns related to the language/tutorial content of the manuscript, as well as the description of the experimental setup and the inferred results. Specifically, the public availability of the test images involved in this study and the sensitivity of the proposed algorithm when applied to more general scenarios need to be made clear. For instance, Table 1 provides a highly specific environment for the conducted tests. How would the proposed technique generalize to other platforms? Moreover, the reviewers have pointed out the absence of comparative results with state-of-the-art works. While this exact problem may not have been addressed in the literature, it is prudent to provide an ablative study to provide quantitative measures to justify the individual components of the proposed algorithm depicted in Figure 2.   

We look forward to receiving your revised manuscript.

Kind regards,

Muhammad Bilal, Ph.D.

Academic Editor

PLOS ONE

Journal Requirements:

"This work was supported by the Shaanxi Provincial Department of Education’s General Special Scientific Research Project (23JK0417), the Shangluo University Natural Science Research Project (22SKY002), and the Shaanxi Provincial Education Science Planning Project (SGH23Y2582)."

"This work was supported by the Shaanxi Provincial Department of Education’s General Special Scientific Research Project (23JK0417), the Shangluo University Natural Science Research Project (22SKY002), and the Shaanxi Provincial Education Science Planning Project (SGH23Y2582)."

"This work was supported by the Shaanxi Provincial Department of Education’s General Special Scientific Research Project"

5. We note that your Data Availability Statement is currently as follows: All relevant data are within the manuscript and its Supporting Information files.

Reviewers' comments:

Reviewer's Responses to Questions

**Comments to the Author**

1. Is the manuscript technically sound, and do the data support the conclusions?

Reviewer #1: Yes

Reviewer #2: Partly

Reviewer #3: Partly

Reviewer #4: Partly

2. Has the statistical analysis been performed appropriately and rigorously? 

Reviewer #1: No

Reviewer #2: No

Reviewer #3: I Don't Know

Reviewer #4: No

3. Have the authors made all data underlying the findings in their manuscript fully available?

Reviewer #1: Yes

Reviewer #2: No

Reviewer #3: Yes

Reviewer #4: No

4. Is the manuscript presented in an intelligible fashion and written in standard English?

Reviewer #1: Yes

Reviewer #2: Yes

Reviewer #3: No

Reviewer #4: No

5. Review Comments to the Author

Reviewer #1: The manuscript proposes "Plasma Arc Flame Length Detection Technology Based on Region of Interest."

The reviewer's concerns are as follows:

1- The novelty of this study is not clear. A flowchart summarizing what was done should be added to the article.

2- Authors have not provided comparison between their proposed methodology and the previous ones in the introduction. According to the current introduction, there is no contribution in the current study.

3- Some figures have poor quality. The quality of the figures should be improved.

4- Reference numbers should be increased.

Reviewer #2: The authors worked on interesting research with a design of region of interest (ROI) based on image detection as well as correction function of the system. Authors further made effort to enhance the real-time performance and accuracy of detection.

However, should try to rewrite the result section of the abstract to reflect more results that complements the technical achievements of the research beyond excerpt error margin that was stated in the abstract section.

Keywords should be revised and sentenced should be removed and authors should only formulate keywords and not key sentences.

The authors approach in the determination of the focal length and selection of lens based on detection requirements of the plasma arc flame is highly commendable.

Authors may revisit the detection system vis-à-vis the overall architectural design. Examining the components of the detection system flow which include noise filtering, threshold segmentation etc. It can be observed that these components do not accurately fit into the section indicated in the overall architecture rather the data acquisition section hence the need for more clarification in this regard so as not to confuse the readers.

Formulas may be referred to as equations and should be mentioned before the place of formulas for proper flow of reading and citing.

Since authors emphasis is on the region of interest, it will be of a great value addition to this research to indicate dimensions and technical information of the specific area of interest in the diagrams of figures 4 and 5 in addition to the threshold. Showing the precise location and dimension of interest will improve the detection mechanism precisely. This will complement authors information on distance analysis from the camera to the plasma arc. Showcasing these distances from the previous illustrations for the impact on detection accuracy, before figure 6 is very important.

Overall a good concept but the results still require a significant level of improvement to indicate how accurate and comparison with previous work done in this area is completely missing from this work.

Grammar should be checked for proper construction as well

Reviewer #3: Please correct grammar and English. I.e. in the Abstract part there are some phrases that are not clear: the accuracy is better than the existing system detection I think not than the actual length of the plasma arc, because you mentioned 1mm, this should be the accuracy for the length but your idea is not well expressed. Before figure 7 you mention that the accuracy depends of the subjective consciousness. This is very interesting and should me more details about it. How did you figure it? To have a better impact, your manuscript should present also a comparison with existing method and how you can state your novel method accuracy. The manuscript must be improved.

Reviewer #4: 1. At the end of the "Introduction" section, the organization and structure of the article should be written

2. The abstract of the paper should be improved by adding your findings.

3. The literature survey is very weak ; the critical survey must be done thoroughly by the authors to improve the paper and also add new subsection under the introduction section

4. The research gap should be written clearly at the end of the introduction section from the literature survey

5.The conclusion section and/or experimental results section must be improved by integrating more generated results or rather stress more description on the results is necessary.

6.There are few editorial and grammatical errors in some parts of the manuscript which can be identified easily by a careful reading. The authors can take help of native English speaker for corrections.

6. PLOS authors have the option to publish the peer review history of their article (what does this mean? ). If published, this will include your full peer review and any attached files.

**Do you want your identity to be public for this peer review?** For information about this choice, including consent withdrawal, please see our Privacy Policy .

Reviewer #1: No

Reviewer #2: No

Reviewer #3: No

Reviewer #4: No

---

## [Author Response · Author response to Decision Letter 1]

3 Dec 2024

List of Responses

Dear Editors and Reviewers,

Thank you for giving us the opportunity to submit a revised draft of the manuscript “PONE-D-24-07571” for publication in the PLOS ONE. We appreciate the time and effort that you and the reviewers dedicated to providing feedback on our manuscript and are grateful for the insightful comments on and valuable improvements to our paper. We have incorporated most of the suggestions made by the reviewers. Those changes are highlighted in the manuscript. Please see below, in blue, for a point-by-point response to the reviewers’ comments and concerns. All page numbers refer to the revised manuscript file with tracked changes.

Reviewer #1

1.The novelty of this study is not clear. A flowchart summarizing what was done should be added to the article.

Response to comment: In the revised manuscript, We have added a flowchart to represent what we have done, please read the introduction section for details.

The novelty of this research: 1. We designed a metal powder device based on the detection of plasma arc length, changing the original control of the rod feed to control the feed of the plasma gun. By detecting the plasma arc length, we can control the feed speed of the servo feeding mechanism, keeping the distance between the metal rod end and the plasma gun constant. This is beneficial for improving the quality of the prepared metal powder and avoids the occurrence of safety issues. 2. We proposed a plasma arc length detection technology based on the region of interest, applying image processing technology to the detection of plasma arc length. We designed a plasma arc detection algorithm and a correction algorithm based on the region of interest, achieving real-time detection of the plasma arc length. This detection method does not affect the plasma arc itself and does not require placing detection devices into the plasma arc, which is of great significance for improving the safety and reliability of plasma detection.

Figure 1. Metal pulverizing device based on detection of plasma arc flame length detection

Figure 1 shows the reason why we do the plasma arc flame length detection and what we do. In the original metal pulverizing process, plasma gun injection plasma arc flame burning metal rod, metal bar end surface under the action of high-temperature plasma arc melting and the action of centrifugal force droplets, then condensation for metal powder, with the metal rod melting, servo mechanism drive metal bar transverse feed, due to the plasma arc is very unstable, without this pulverizing process cannot guarantee the metal bar end surface and plasma arc plasma gun distance constant, will affect the quality of the metal powder, and also can cause some safety accidents. Therefore, we designed the metal powder device based on the plasma arc flame length detection, the original control rod feed design to control the plasma gun feed, by detecting the plasma arc flame length to control the servo input speed, makes the metal rod end surface and plasma gun distance between constant, is beneficial to improve the quality of the metal prepared powder and avoid the occurrence of safety problems. As a special flame, a plasma arc has the characteristics of high temperature and high-intensity light, and it is difficult to detect its length by ordinary methods, to realize the measurement of plasma arc length, This study was presented based on the interest of Plasma arc flame length detection technology, will be the image processing technology Application in plasma arc flame length detection, design based on the region of interest of plasma arc flame detection algorithm and correction algorithm, realize the real-time detection of plasma arc flame length, this detection method will not affect the plasma arc itself, also need not probe into plasma arc flame, to improve the safety and reliability of the plasma detection is of great significance.

2.Authors have not provided comparison between their proposed methodology and the previous ones in the introduction. According to the current introduction, there is no contribution in the current study.

Response to comment: In the revised manuscript, We have added some references regarding plasma detection. Please read the introduction section for details.

As shown in Figure S2, after locating the position of the plasma arc flame in the image, this study designs a plasma arc flame length detection system, which can accurately and real-time measure the length information of the plasma arc flame.The system is mainly used for real-time monitoring of the plasma arc flame on-site, displaying the image of the plasma arc flame and the length of the plasma arc flame.

Figure S2. Plasma arc flame length detection system

As shown in Figures S3, Since the servo feeding mechanism is controlled by KingView software, after the detection system detects the length of the plasma arc flame, it needs to transmit the detected data and some parameter contents to KingView for display. As shown in Figures S3, we designed and implemented the KingView display interface. The left side of the interface displays the original image and the processed image of the plasma arc flame, while the right side shows the detection parameters and the length of the plasma arc flame. The identification area and position area and coordinates are settings for the region of interest. In the experiment, the position is set according to the area of the plasma arc flame in the image. The length and width of the region of interest cannot be set too large or too small; too large will affect the real-time detection, and too small will fail to detect the complete plasma arc flame.

Figure S3. Plasma arc flame detection and monitoring center system based on Kingview

3.Some figures have poor quality. The quality of the figures should be improved.

Response to comment: We have changed figure 2 and figure 3.If there are any other modifications we could make, we would like very much to modify them and we really appreciate your help.

Figure 2. Overall Architecture Design of Plasma Arc Flame Detection System.

Figure 3. Detection system process

4.Reference numbers should be increased.

Response to comment:We have added some references.

14. Department Of Physics, York Plasma Institute University, York Plasma Institute University Department Of Physics, York Plasma Institute University Department Of Physics, York Plasma Institute University Department Of Physics, and York Plasma Institute University Department Of Physics. "Plasma Temperature Measurements Using Black-Body Radiation From Spectral Lines Emitted by a Capillary Discharge." Journal of Quantitative Spectroscopy and Radiative Transfer 220 (2018): 1-04.

15. Prasad, Dandu, Gusarov Andrei, Moreau Philippe, Leysen Willem, Kim SungMoon, Mégret Patrice, and Wuilpart Marc. "Plasma Current Measurement in Iter with a Polarization-Otdr: Impact of Fiber Bending and Twisting On the Measurement Accuracy." APPLIED OPTICS 61, no. 9 (2022): 2406-16.

16. Zhu, Jiajian, Jinlong Gao, Andreas Ehn, Marcus Alden, Zhongshan Li, Dmitry Moseev, Yukihiro Kusano, Mirko Salewski, Andreas Alpers, Peter Gritzmann, and Martin Schwenk. "Measurements of 3D Slip Velocities and Plasma Column Lengths of a Gliding Arc Discharge." APPLIED PHYSICS LETTERS 106, no. 4 (2015): 44101.

Reviewer #2

1.The authors worked on interesting research with a design of region of interest (ROI) based on image detection as well as correction function of the system. Authors further made effort to enhance the real-time performance and accuracy of detection.

However, should try to rewrite the result section of the abstract to reflect more results that complements the technical achievements of the research beyond excerpt error margin that was stated in the abstract section.

Response to comment: Thank you very much for your compliment. In the revised manuscript, We have already rewritten the abstract section, adding a description of the results, please read the abstract for details.

Here is our rewritten Abstract: With the rapid development of metal 3D printing technology, the demand for spherical metal powder for 3D printing materials is increasing day by day, and the technology of high-quality spherical metal powder has become the key research content of many enterprises and research institutions in various countries. Traditional plasma rotary electrode legal powder technology of servo into the feed speed is artificially set, can not guarantee the end of the powder and plasma gun distance constant, easy to lead to the quality of the metal powder is affected and there is a large safety hazard, for these problems, this paper designed based on the area of interest, the image processing technology applied to the plasma arc flame length detection, through the introduction of image detection of the area of interest and arc flame length correction function, improve the real-time detection and detection accuracy, Finally, the real-time monitoring of the detection site is realized through the configuration king. The experimental results show that the edge of the image target area is smooth, has clear brightness, and no noise, which can meet the requirements of subsequent image processing and monitoring requirements. After correction, the plasma arc flame length is about 40 mm. Compared with the plasma arc flame length to be controlled, the detection error is less than 1 mm, and the length change is relatively stable to meet the measurement requirements. As changes over time, the length of the plasma arc flame changes little, and the distance between the plasma gun and the metal rod end surface with the passage of time and metal rod melting did not change significantly, the controller can according to the detection of the plasma arc flame length control the servo feed speed, makes the plasma arc flame in the process of pulverizing distance constant, meet the actual industrial production needs.

2.Keywords should be revised and sentenced should be removed and authors should only formulate keywords and not key sentences.

The authors approach in the determination of the focal length and selection of lens based on detection requirements of the plasma arc flame is highly commendable.

Response to comment: In the revised manuscript, We have modified the keywords.

Keywords: Plasma detection, Length detection, Region of interest

3.Authors may revisit the detection system vis-à-vis the overall architectural design. Examining the components of the detection system flow which include noise filtering, threshold segmentation etc. It can be observed that these components do not accurately fit into the section indicated in the overall architecture rather the data acquisition section hence the need for more clarification in this regard so as not to confuse the readers.

Response to comment: In the revised manuscript, We have clarified the parts indicated by these components: The contents described in the detection system process in FIG. 3 are the contents of the image processing and measurement software in FIG. 2. After the action of the acquisition system in FIG. 2, the color image of the plasma arc flame is acquired. As shown in FIG. 3, the color image is usually converted into a grayscale image first, Then the grayscale images are processed.

4.Formulas may be referred to as equations and should be mentioned before the place of formulas for proper flow of reading and citing.

Response to comment：In the revised manuscript, We have referenced the formula before its appearance.

5.Since the authors' emphasis is on the region of interest, it will be of great value addition to this research to indicate dimensions and technical information of the specific area of interest in the diagrams of figures 4 and 5 in addition to the threshold. Showing the precise location and dimension of interest will improve the detection mechanism precisely. This will complement the authors information on distance analysis from the camera to the plasma arc. Showcasing these distances from the previous illustrations for the impact on detection accuracy, before Figure 6 is very important.

Response to comment:In the revised manuscript,We have explained the size information of the region of interest：Generally, we set the area of interest for the image of the size of 500 * 500 pixels。

6.Overall a good concept but the results still require a significant level of improvement to indicate how accurate and comparison with previous work done in this area is completely missing from this work.

Response to comment:We have placed some results in the supplementary materials; please refer to the supplementary materials for more information on our experimental results.

As shown in Figure S2, after locating the position of the plasma arc flame in the image, this study designs a plasma arc flame length detection system, which can accurately and real-time measure the length information of the plasma arc flame.The system is mainly used for real-time monitoring of the plasma arc flame on-site, displaying the image of the plasma arc flame and the length of the plasma arc flame.

Figure S2. Plasma arc flame length detection system

As shown in Figures S3, Since the servo feeding mechanism is controlled by KingView software, after the detection system detects the length of the plasma arc flame, it needs to transmit the detected data and some parameter contents to KingView for display. As shown in Figures S3, we designed and implemented the KingView display interface. The left side of the interface displays the original image and the processed image of the plasma arc flame, while the right side shows the detection parameters and the length of the plasma arc flame. The identification area and position area and coordinates are settings for the region of interest. In the experiment, the position is set according to the area of the plasma arc flame in the image. The length and width of the region of interest cannot be set too large or too small; too large will affect the real-time detection, and too small will fail to detect the complete plasma arc flame.

Figure S3. Plasma arc flame detection and monitoring center system based on Kingview

7.Grammar should be checked for proper construction as well

Response to comment:In the revised manuscript,we have corrected some grammatical issues.

Reviewer #3

1.Please correct grammar and English.

Response to comment:In the revised manuscript, we have corrected some grammatical issues.

2.I.e. in the Abstract part, there are some phrases that are not clear: the accuracy is better than the existing system detection I think not than the actual length of the plasma arc, because you mentioned 1mm, this should be the accuracy for the length but your idea is not well expressed.

Response to comment:We have rephrased these unclear sentences in the summary:After correction, the plasma arc flame length is about 40 mm. Compared with the plasma arc flame length to be controlled, the detection error is less than 1 mm, and the length change is relatively stable to meet the measurement requirements.

3.Before figure 7 you mention that the accuracy depends of the subjective consciousness. This is very interesting and should me more details about it. How did you figure it?

Response to comment:Due to the angle of the camera capturing the plasma arc flame being influenced by subjective human factors, the factors affecting the detection accuracy are significantly influenced by subjective consciousness. Therefore, we designed a correction algorithm to adjust the detection results.

4.To have a better impact, your manuscript should present also a comparison with existing method and how you can state your novel method accuracy. The manuscript must be improved.

Response to comment:In the revised manuscript,We have made improvements to the manuscript and added some references regarding plasma detection. Please read the introduction section for details.

The detection of plasma arcs has also been studied by some scholars.document(14)Using spectroscopy to measure the plasma arc, in which blackbody radiation can be applied to extremely strong lines and can be used to measure the temperature of the plasma arc. Prasad class(15)A simulation-based method was developed to measure plasma current in t

---

## [Decision Letter · Decision Letter 1]

1 Jan 2025

PONE-D-24-07571R1Research on Plasma Arc Flame Length Detection Technology Based on Region of InterestPLOS ONE

Dear Dr. Jiang,

Thank you for submitting your manuscript to PLOS ONE. After careful consideration, we feel that it has merit but does not fully meet PLOS ONE’s publication criteria as it currently stands. Therefore, we invite you to submit a revised version of the manuscript that addresses the points raised during the review process.

We look forward to receiving your revised manuscript.

Kind regards,

Muhammad Bilal, Ph.D.

Academic Editor

PLOS ONE

Additional Editor Comments:

The authors have made some effort in improving the quality of the manuscript. However, it still has several shortcomings. For instance,

1- The language and tutorial need to be improved. It is suggested to seek help from a native English speaker or assistance from a formal proofreading service. For instance,

a. In the abstract, the following sentence describes the main problem and is yet hard to read: “Traditional plasma rotary electrode legal powder technology of servo into the feed speed is artificially set, can not guarantee the end of the powder and plasma gun distance constant, easy to lead to the quality of the metal powder is affected and there is a large safety hazard, for these problems,”

b. The commas and periods need to be appropriately used throughout the manuscript.

c. A few reference works have been cited using square brackets “[]” while for others, round brackets “()” have been used. Please use a consistent form according to the PLOS ONE suggested referencing style.

d. Please elaborate what is “configuration king”

2- While discussing figures 7 and 8, please explain how the “actual length” of the plasma arc was measured? While discussing equations 1 to 5, a manual measurement method using a ruler has been described. Clearly, this method cannot be used while the plasma arc is actually in operation.

3- Time axis in figure 9 has the units of “mm”. Please explain

Reviewers' comments:

Reviewer's Responses to Questions

**Comments to the Author**

1. If the authors have adequately addressed your comments raised in a previous round of review and you feel that this manuscript is now acceptable for publication, you may indicate that here to bypass the “Comments to the Author” section, enter your conflict of interest statement in the “Confidential to Editor” section, and submit your "Accept" recommendation.

Reviewer #3: All comments have been addressed

Reviewer #4: All comments have been addressed

2. Is the manuscript technically sound, and do the data support the conclusions?

Reviewer #3: Yes

Reviewer #4: Yes

3. Has the statistical analysis been performed appropriately and rigorously? 

Reviewer #3: I Don't Know

Reviewer #4: Yes

4. Have the authors made all data underlying the findings in their manuscript fully available?

Reviewer #3: Yes

Reviewer #4: Yes

5. Is the manuscript presented in an intelligible fashion and written in standard English?

Reviewer #3: Yes

Reviewer #4: Yes

6. Review Comments to the Author

Reviewer #3: Authors have addressed all the comments of the reviewers. The manuscript is improved presenting clearly the outcomes of the research.

Reviewer #4: Thank you for addressing my comments and improving your manuscript. The authors answered correctly to all my comments.

In my opinion the article can be accepted for publication.

7. PLOS authors have the option to publish the peer review history of their article (what does this mean? ). If published, this will include your full peer review and any attached files.

**Do you want your identity to be public for this peer review?** For information about this choice, including consent withdrawal, please see our Privacy Policy .

Reviewer #3: No

Reviewer #4: No

---

## [Author Response · Author response to Decision Letter 2]

15 Feb 2025

List of Responses

Dear Editors and Reviewers,

Thank you for giving us the opportunity to submit a revised draft of the manuscript “PONE-D-24-07571” for publication in the PLOS ONE. We appreciate the time and effort that you and the reviewers dedicated to providing feedback on our manuscript and are grateful for the insightful comments on and valuable improvements to our paper. We have incorporated most of the suggestions made by the reviewers. Those changes are highlighted in the manuscript. Please see below, in blue, for a point-by-point response to the reviewers’ comments and concerns. All page numbers refer to the revised manuscript file with tracked changes.

Editors

1.The language and tutorial need to be improved. It is suggested to seek help from a native English speaker or assistance from a formal proofreading service. For instance,

a.In the abstract, the following sentence describes the main problem and is yet hard to read: “Traditional plasma rotary electrode legal powder technology of servo into the feed speed is artificially set, can not guarantee the end of the powder and plasma gun distance constant, easy to lead to the quality of the metal powder is affected and there is a large safety hazard, for these problems,”

Response to comment: In the revised manuscript, we have modified this section, please read the abstract section for details.

Abstract: With the rapid advancement of metal 3D printing technology, there is a growing demand for spherical metal powder as a primary material for 3D printing. The process technology that ensures the production of high-quality spherical metal powder has become a focal area of research for numerous enterprises and research institutions globally. In the conventional plasma rotating electrode method for powder production, the feed speed of the servo feeding mechanism is manually predetermined, leading to potential variations in the distance between the end face of the metal rod and the plasma gun that generates the plasma arc. Such inconsistency can compromise the quality of the metal powder produced and pose safety hazards if the gap between the metal rod and the plasma gun is too narrow. To address these issues, this study presents a novel plasma arc length detection system based on the concept of the region of interest. The proposed system leverages image processing technology for efficiently detecting the plasma arc length. By incorporating image detection within the region of interest alongside an arc length correction function, the system enhances real-time performance and detection precision. Additionally, real-time monitoring of the detection site is enabled through KingView. Experimental findings indicate that the image target area post plasma arc detection exhibits well-defined edges, clear brightness, and minimal noise, thereby meeting the prerequisites for subsequent image processing and monitoring tasks. The corrected plasma arc length averages around 40mm, with a detection error of less than 1mm when compared to the desired controlled plasma arc length. Moreover, the length variation remains relatively stable, thus fulfilling the measurement criteria. Over time, the detected plasma arc length exhibits negligible fluctuations, suggesting consistent proximity between the plasma gun and the end face of the metal rod during the melting process. The controller can dynamically control the feed speed of the servo feeding mechanism according to the detected plasma arc length, ensuring a constant distance between the plasma arc and the end face of the metal rod throughout the powder production process, thus aligning with practical industrial requirements.

b. The commas and periods need to be appropriately used throughout the manuscript.

Response to comment:Thank you very much for your suggestion. In the latest manuscript, we have carefully reviewed the use of commas and periods and made modifications in the latest manuscript.Please read the follwing:

Muhammad et al. [8] presented a flame identification method utilizing the GoogleNet model [9].

During this process, precise control of the plasma arc is essential as any fluctuations can jeopardize the consistency of the metal powder and potentially lead to safety hazards. To address these challenges, we have devised a metal powder device that incorporates a plasma arc flame length detection system.

The first section is the Introduction, which provides a concise overview of the content and significance of the paper.

Lens selection is contingent upon magnification and focal length requirements. The desired lens magnification should adhere to the formula: camera pixel count * vertical resolution / detection field of view = 0.12. Additionally, the focal length of the lens is determined by: working distance * (magnification / 1 + magnification) = 134 millimeters. A tabulated overview of the acquisition system components is presented in Table 1 for reference.

To carry out this procedure, a ruler is positioned adjacently to the plasma arc flame, while maintaining a constant camera angle. Subsequently, an image of the ruler is captured and subjected to the same pre-processing steps detailed previously.

The identified target area's boundary within the plasma arc flame appears well-defined with a smooth edge, exhibiting a clear and noise-free brightness level.

The identified target area's boundary within the plasma arc flame appears well-defined with a smooth edge, exhibiting a clear and noise-free brightness level. This ensures suitability for requisite image processing and monitoring tasks.

Figure 8 portrays the corrected results, showcasing the efficacy of the formula in refining the detected length.

As can be seen from Figure 7, the corrected plasma arc flame length is about 40mm.

Figure 9 illustrates the lengths of the plasma arc flames detected during three separate intervals within the same experimental setup, each spaced 20 minutes apart.

c. A few reference works have been cited using square brackets “[]” while for others, round brackets “()” have been used. Please use a consistent form according to the PLOS ONE suggested referencing style.

Response to comment:Thank you very much for your correction. In the revised manuscript, we have followed the citation style recommended by PLOS ONE and uniformly cited references using square brackets “[]”. Please read the follwing:

As the global manufacturing sector upgrades, manufacturing is growing faster than ever. Notably, 3D printing technology [1] stands as a representative of advanced manufacturing techniques that have significantly impacted the development of the manufacturing industry. In the realm of 3D printing, metal powder serves as a key material [2]. The main methods for preparing metal powders include plasma rotating electrode preparation (PREP), plasma atomization (PA), and aerosol gas atomization (GA) techniques [3]. Among these methods, the plasma rotating electrode approach is highlighted for its ability to produce spherical metal powder with minimal oxygen content and high cleanliness [4]. In the plasma rotating electrode method, the process of preparing metal powders is typically carried out in a controlled gas environment of high-purity argon to prevent oxidation and impurities. Despite its advantages in producing high-quality metal powders, challenges such as empirical setting of the metal rod feed speed and the instability of the plasma arc can impact the consistency and safety of the process. Maintaining a constant distance between the metal rod and the plasma gun to ensure optimal powder quality is crucial, as deviations in this distance can lead to safety hazards and operational issues, such as explosions or back spray incidents.

Extensive research has been dedicated to enhancing flame detection technologies. Jiang B et al. [5] proposed a method that combines local texture features and global color features of the LAB histogram to achieve swift and accurate flame detection. Similarly, another study [6] utilizes a classifier to pinpoint the flame's target area through the extraction of texture features in the flame region. With the rapid evolution of deep learning technologies, neural networks have also found application in the sphere of flame detection. Frizzi et al. [7] introduced a flame detection approach based on convolutional neural networks, which exhibits effective detection capabilities; however, the detection accuracy varies significantly when encountering flames of different colors. Muhammad et al. [8] presented a flame identification method utilizing the GoogleNet model [9]. This method incorporates transfer learning to fine-tune model parameters, striking a balance between detection efficiency and accuracy, albeit resulting in a reduction in detection accuracy. In a separate document [10], a video-based flame recognition method was proposed. By employing multi-scale fusion to extract flame features, the detection rate was enhanced, significantly boosting real-time performance. Furthermore, a study [11] introduced a flame identification algorithm based on an enhanced version of the YOLOv5 network [12]. Leveraging transfer learning [13], this method enhances the small target detection layer, augments the model's correlation identification capacity, and ultimately improves flame detection accuracy. While these methods have shown promise in detecting parameters such as flame shape, category, and area, they fall short in capturing length information of the flame.

Some researchers have explored the detection of plasma arc in various studies. A noteworthy piece of literature [14] employs spectroscopy as a technique to assess plasma arc, utilizing blackbody radiation to analyze prominent spectral lines and determine the temperature of the plasma arc. Prasad et al. [15] introduced a simulation-based approach to quantify the plasma current within the International Thermonuclear Experimental Reactor (ITER). Their method involved investigating the polarization rotation induced by Faraday effect in a rotating fiber optic sensor situated around the vacuum chamber, accounting for bending and twisting effects to assess the reflectometer's performance in gauging plasma current in the ITER facility. Another scholarly work [16] utilized a pair of synchronized high-speed cameras to capture the dynamics of both the plasma column and tracer particles. This study conducted a comprehensive 3D data analysis of the column and tracer particles, encompassing the reconstruction of the plasma column in three dimensions and the measurement of tracer particles' velocities using discrete tomography. By determining the 3D slip velocity and length of the plasma column, researchers were able to accurately estimate the radius of the conductive zone within the plasma column. While existing research has focused on measuring parameters such as plasma arc temperature, density, and conductivity, investigations specifically targeting the measurement of plasma arc length remain limited.

The demand for detecting plasma arc flame length in this study comprises several crucial components. Firstly, there is a need to capture the video signal of the plasma arc in real-time and accurately determine the precise position and length of the plasma arc flame. Subsequently, the identified video signal and length of the plasma arc flame should be promptly transmitted to the KingView software for the servo feed system to access the information instantaneously [19]. Finally, these data must be presented in the KingView interface with the flexibility to adjust detection parameters such as binary parameters, brightness, and filter width [20]. Consequently, the holistic framework of the plasma arc flame detection system is structured into three main segments: the acquisition system, detection system, and display system, illustrated in Figure 2.

The processes outlined in the detection system depicted in FIG. 3 correspond to the functionalities of the image processing and measurement software featured in FIG. 2. Subsequent to the actions of the acquisition system in FIG. 2, the system secures a color image of the plasma arc flame. As indicated in FIG. 3, the color image is typically initial transitioned into a grayscale format [21], following which the grayscale images undergo processing. Despite passing through the capture card, the images captured by industrial cameras often retain a considerable amount of noise [22]. Removing these noise elements is imperative for the detection goals. In this study, the approach involves employing median filtering [23] to eliminate noise within the image. Median filtering proves highly effective in eliminating discrete noise points within the image. For each given area, the algorithm selects a pixel point within that region. The median pixel value is then designated as the value of the central point, effectively eliminating point noise and preserving the sharpness of image edges, thereby enhancing overall image fidelity. To detect the plasma arc flame in the collected image, it is necessary to remove the background, extract the target flame area, and perform threshold segmentation on the image [24] to separate the target area from the background. Threshold segmentation involves determining a suitable threshold level to classify each image point as either background or target, hence delineating the target region. The threshold determination methodology is experimentally based, with the study ultimately selecting a threshold value of 200 after numerous trials to achieve optimal segmentation results. As the plasma arc flame tends to emit high brightness, small holes may persist post-segmentation. Addressing these holes is crucial to accurate plasma arc flame length calculations. Leveraging morphological operations [25], these holes can be effectively filtered out. Through multiple experiments, the study opts for a 5*5 square structural element for opening the image post-threshold segmentation. This method involves erosion followed by dilation, preserving the target area's shape while removing smaller holes without compromising the image condition. Complying with the plasma arc flame detection requirements necessitates extracting the edges within the target area. Consequently, conducting edge detection on the target area is pivotal. This study uses the canny algorithm [26] for edge detection, utilizing dual thresholds [27] to ascertain genuine and potential edges within the target region. Notably, the advantage of this algorithm is its resistance to noise interference, with the extracted edges demonstrating impressive continuity while avoiding false positives or edge loss.

d. Please elaborate what is “configuration king”

Response to comment:Thank you very much for your correction, "configuration king" was a translation error, the correct term should be "KingView". In the revised manuscript, We have modified "configuration king" to "KingView".

KingView is a comprehensive and user-friendly industrial automation monitoring system software that can collect real-time operational data from various industrial devices, such as temperature, pressure, flow, etc., and display and monitor them through intuitive graphical interfaces. With its comprehensive features and robust performance, KingView plays a significant role in the field of industrial automation, providing engineers with efficient and reliable monitoring and management tools. In this article, as shown in Figure S1, we designed and implemented the KingView display interface. The left side of the interface presents the original image of the plasma arc flame and its processed image, while the right side details the detection parameters and the detection length of the plasma arc flame.

Figure S1. Plasma arc flame detection and monitoring center system based on Kingview

2.While discussing figures 7 and 8, please explain how the “actual length” of the plasma arc was measured? While discussing equations 1 to 5, a manual measurement method using a ruler has been described. Clearly, this method cannot be used while the plasma arc is actually in operation.

Response to comment:As shown in Figure 1, the “actual length” of the plasma arc flame refers to the set distance “d” between the plasma gun and the metal rod material. Before starting the m

---

## [Editor Report · Decision Letter 2]

3 Mar 2025

Research on Plasma Arc Flame Length Detection Technology Based on Region of Interest

PONE-D-24-07571R2

Dear Dr. Jiang,

We’re pleased to inform you that your manuscript has been judged scientifically suitable for publication and will be formally accepted for publication once it meets all outstanding technical requirements.

Kind regards,

Muhammad Bilal, Ph.D.

Academic Editor

PLOS ONE
---

## [Editor Report · Acceptance letter]

PONE-D-24-07571R2

PLOS ONE

Dear Dr. Jiang,

I'm pleased to inform you that your manuscript has been deemed suitable for publication in PLOS ONE. Congratulations! Your manuscript is now being handed over to our production team.

Kind regards,

on behalf of

Dr. Muhammad Bilal

Academic Editor

PLOS ONE